# Experimental Study on Foci Development in Mortar Using Seawater and Sand

**DOI:** 10.3390/ma12111799

**Published:** 2019-06-03

**Authors:** Yajun WANG, Chuhan ZHANG, Jinting WANG, Yanjie XU, Feng JIN, Youbo WANG, Qian YAN, Tao LIU, Xiaoqing GAN, Zhan XIONG

**Affiliations:** 1Zhejiang Key Laboratory of Offshore Marine Engineering Technology, Zhoushan 316002, China; 2School of Port and Transportation Engineering, Zhejiang Ocean University, Zhoushan 316002, China; 18227822920@163.com (Q.Y.); 13071795013@163.com (T.L.); 18368090212@139.com (Z.X.); 3State Key Laboratory of Hydroscience and Hydraulic Engineering, Tsinghua University, Peking 100084, China; wangjt@tsinghua.edu.cn (J.W.); xuyanjie@tsinghua.edu.cn (Y.X.); jinfeng@tsinghua.edu.cn (F.J.); 4Yangtze River Scientific Research Institute, Wuhan 430010, China; gxqxf@sina.com

**Keywords:** mortar, seawater and sand, interfacial behavior, strength, damage, on-line detection

## Abstract

Mortar using seawater and sand was the material studied here. The mortar specimens, in particular, were cured in natural seawater. The foci development in the mortar was the principal interest in this study. The on-line damage detection experiment art, including dynamically global MSHCT (Multi-Slices Helical Computer Tomography) scan and the local detection of EDS (Energy Dispersive Spectrometer), SEM (Scanning Electron Microscopy), and XRD (X-ray Diffraction) was designed to research the foci development in the mortar specimen. The mortar specimens with 70-day age were produced and investigated by the on-line damage detection experiments. The experiment results indicated that the mortar using seawater and sand offered appreciable strength at the early age, at least, although some saline minerals were generated during the preparation. The residual strength of the mortar was above 13 MPa, which helped to prevent the sharp damage of engineering bodies. The micro-interfacial behavior and the parental foci development controlled the damage evolution in the mortar using seawater and sand, the performance of which was still the adjustable one by composition optimization.

## 1. Introduction

Nowadays, the cement-base materials play the predominant role in most infrastructures and even military construction [1], and the application of water and sand beyond question is the decisive approach in the production of the cement-base materials. The global sand consumption nowadays reaches 45 billion metric tons per year [2]. Meantime, the worldwide resource of the fluvial-lacustrine sand is being exhausted (Figure 1). 

Moreover, the fresh water also shows the growing discrepancy between the shrinking resource and the expanding consumption, and the negative effect of the fresh water shortage has aggravated the cement-base materials application, in particular, in the marine environment [4].

The development of the instant composites (including the instant cement-base ones) has been the crucial duty, especially in the marine environment. Seawater and sand are the key constituents for the development of the instant cement-base composites in the offshore construction [5,6,7,8]. The seawater curing process is the preferable choice for the cement-base composites preparation in the marine environment as the result of the instant efficiency [9,10,11]. Wang et al. [12] carried out multi-scale investigations on hydration mechanisms of ordinary Portland cement with seawater mixing and reported that the seawater cement paste demonstrates a highly stiffened structure for its rheological behavior at the fresh stage. Their work also mentioned that the formation of Friedel’s salt encourages future studies on chemistry between seawater and cements to beneficially utilize the seawater ingredients to improve the material properties and structural durability. Shi et al. [13] conducted the chloride ingress experiment to measure the Friedel’ salt profiles in the mortar samples, and their results reported that the maximum amount of Friedel’s salt was found in the region with limited leaching of calcium. Yu et al. [14] carried out the mechanical tests to study the effect of seawater mixing on the properties of potassium magnesium phosphate cement paste and reported that adding some mineral admixtures into the paste that was prepared with seawater could lessen the negative effect of seawater for mixing on the properties of hardened body. Similarly, Yang et al. [15] proposed the addition of limestone powder and silica fume in the magnesium potassium phosphate cement paste the early-age strength of which could be improved in seawater. Yang et al. [16] also mentioned the effect of the early seawater curing on the properties of magnesium potassium phosphate cement paste, and their results indicated that the performance of the paste that was cured in seawater after three-day natural curing showed clear improvement, thanks to the higher degree of crystallinity and more perfect pore size distribution. Li et al. [17] conducted the thermal experiments to investigate the properties of alkali-activated slag paste, mortar and concrete utilizing seawater and sea sand, and offered the conclusion that the materials degradation is caused by thermal mismatch between the paste matrix and aggregates, regardless of the use of cement or slag, freshwater or seawater, and river sand or sea sand. The work of Wang et al. [18] reported that metakaolin addition improved the hydration, mechanical properties, and durability of the coral waste mortar. The materials production in their work can relieve the lack of locally available concrete ingredients on isolated islands. Tan et al. [19] also produced the coral aggregate seawater concrete and investigated its strength by comparison with the ones of ordinary Portland concrete and lightweight aggregate concrete. Their results proposed that the aggregate type has significant influence on the testing strength curve of the concrete materials by different methods. Rashad and Ezzat [20] conducted an experimental investigation on the alkali-activated slag pastes that were produced with seawater, etc., and indicated that the higher amount of chloride, sulfate, and sodium ions contained in pastes accelerated the polymerization process and filled its pores by the reaction products and crystallization salts. Especially, Rashad and Ezzat reported that the polymerization process in the seawater pastes enhanced the compressive strength at early ages.

According to the references, it can be drawn that the marine cement-base composites have been the most popular materials in the offshore construction. Erniati et al. [21] reported the importance of seawater application in the concrete production of an archipelago environment where the fresh water was highly of scarcity. Xiao et al. [22] mentioned the tremendous consumption of river sand and freshwater in concrete production, which has raised serious environmental concerns. They also indicated the promising application of sea sand and seawater concrete in many countries, where the construction showed reliability under the proper process. Karthikeyan and Nagarajan [23] ascertained the sea sand usage in concrete construction, which helped to avoid ecological and environmental adversity. Their study revealed that the properly treated sea sand was the acceptable fine aggregate. 

The mortar beyond question is the key composite of the widely used cement-base materials and the rational investigation on mortar performance helps to understand the preparation of the cement-base materials well. Therefore, the on-line damage detection experiment art was designed here to probe the foci development in the mortar specimens, by which, the chemical constituents of the foci were ascertained and the dynamically microscopic deterioration was visualized. Meantime, the strain and stress states (i.e., SSS) of the mortar specimens were quantified to interpret the structural evolution of the foci. The procedure from microscopic deterioration to macro-fragments generation was the foci development that was studied by the on-line damage detection experiment, including dynamically global MSHCT (Multi-Slices Helical Computer Tomography) scan and local detection of EDS (Energy Dispersive Spectrometer), SEM (Scanning Electron Microscopy), and XRD (X-ray Diffraction). Particularly, the experiment art in this study on the foci development was the crucial methodology for the successful utilization of the mortar while using sea waster and sand. Moreover, the mortar using sea waster and sand and cured in natural seawater showed the promising potential. Hence, the study will help to realize the widespread utilization of the cement-base materials using sea waster and sand and cured in natural seawater.

## 2. Materials and Methods

### 2.1. Composition of the Mortar

We conceptualized the study ideology and designed the composition of the mortar (Table 1) in the State Key Laboratory of Hydroscience and Hydraulic Engineering of Tsinghua University (in Peking, China). We produced the mortar specimens in the Zhejiang Key Laboratory of Offshore Marine Engineering Technology of Zhejiang Ocean University (in Zhoushan, China). 

The information on the parental materials was reported according to the composition detail, as follows.

*Seawater:* The mortar specimens were prepared with the seawater that was got from Changzhi Bay of Zhoushan Seas located at 29.98° N and 122.18° E, where the expectation value of pH was 7.86. The acquisition time of the seawater specimen was a.m. 9:30 on Dec 10^th^, 2016 and the quantitative fabric experiment lasted from a.m. 10:30 to p.m. 17:50 on Dec 10^th^, 2016. The quantitative fabric of the seawater specimen, including the key ions and the saline minerals, is shown in Figure 2.

The highest concentration was the one of NaCl with 17997.65 × 10^−3^‰ and the descending sequence of the others was Cl^−^ with 13768.18 × 10^−3^‰, Na^+^ with 7076 × 10^−3^‰, MgCl_2_ with 3574.38 × 10^−3^‰, CaSO_4_ with 1074.74 × 10^−3^‰, Mg^2+^ with 903 × 10^−3^‰, SO_4_^2−^ with 780 × 10^−3^‰, K_2_SO_4_ with 607.66 × 10^−3^‰, Ca^2+^ with 316.1 × 10^−3^‰, K^+^ with 272.4 × 10^−3^‰, and Br^−^ with 40 × 10^−3^‰.

*Sea sand:* The sea sand was excavated from Nansha shore of Zhoushan seas located at 122.43° N and 29.87° E. Table 2 provides the principal properties of the sea sand. 

Particularly, the natural sea sand has been treated by dryer BH-002 (power 4.5 kW, Bohui Ltd., Hangzhou, Zhejiang, China) for 48 hours to thoroughly remove the natural water content for precise material design. 

The size distribution of the sea sand particles was investigated by both normal sieving technology (size range of 0.16 mm ~ 5 mm) and Malvern Mastersizer 2000 analyzer (Malvern Panalytical Ltd., Malvern, England, United Kingdom) (size range of 2 × 10^−^^5^ mm ~ 2mm), and the results are offered in Figure 3. Equations (1) and (2) define the characteristic parameters of the sea sand particles, i.e., the uniformity coefficient *C*_u_ and the curvature coefficient *C*_c_ [21], and their expectation values from the statistical work are reported in Table 3. Moreover, *C*_u_ and *C*_c_ help to evaluate the uniformity and the continuousness of powder materials, inclusive of the sea sand, respectively.
(1)$$C_{u}=\frac{d_{60}}{d_{10}}$$
(2)$$C_{c}=\frac{{\left(d_{30}\right)}^{2}}{d_{60}\cdot d_{10}}$$

Particularly, *d*_10_, *d*_30_, *d*_50_, and *d*_60_ here designate the specific particle sizes with their cumulative composition percentages as 10%, 30%, 50%, and 60%, respectively. 

In terms of uniformity, *C*_u_ with the value beyond 5 indicates that the material particles have the non-uniform distribution. *C*_u_ with the value below 5, by contrast, means that the material particles have uniform distribution. With regard to continuousness, *C*_c_ with the span (1, 3) shows the continuous distribution of the material particles. Otherwise, some intermediate size groups have been lost. Hence, the sea sand of Nansha shore had the uniform and discontinuous distribution.

Meantime, the study carried out the EDS (EDAX Inc., Mahwah, NJ, U.S.A.) and XRD analysis (BRUKER AXS Inc., Madison, WI, U.S.A.) to explore the sea sand composition and Figure 4 shows the results. 

The SEM image of the sea sand is given in Figure 5, where the SiO_2_ base and the original micro-cracks were covered by KAlSi_3_O_8_ texture and Na(Si_3_Al)O_8_ chips. The irregular surface of the sea sand helped to strengthen the connection between the material particle and the hydrate gel. Surely, the chip cover played X factor on the micro-interfacial behavior of the mortar using seawater and sand. Hence, the micro-interfacial behavior of the sea sand and the hydrate gel should be specifically ascertained. 

*Cement:* The mortar was prepared with the composite Portland cement P.C 42.5 R that has been produced by Hailuo Cement Co. Ltd in Southern-Eastern China. The principal properties of the composite Portland cement P.C 42.5 R are reported in Table 4. 

EDS analysis was applied to detect the principal elements’ content of the composite Portland cement P.C 42.5 R and the results are shown in Figure 6, where the values of wt% were the expectations derived from three scanned positions. 

XRD analysis of the composite Portland cement P.C 42.5 R was also implemented and the results reported its principal composition, including Ca_2_SiO_4_, Ca_3_SiO_5_, Ca_3_Al_2_O_6_, SiO_2_, Al_2_O_3_, CaO, and MgO.

### 2.2. Experiments’ Standards, Equipments and Methods

The composition design and the mortar specimens’ preparation mainly obey the Sino standards and partly refer to the American codes [24,25,26,27,28,29].

*Mixing:* The mortar paste was produced in the mixer, the type of which was JJ-5 ISO679-1989(E) (maximal power 0.55 kW and capacity 5 L) produced by Jingwei company of Hebei province, Shijiazhuang, in China. The standard mixing time was five minutes. 

*Molding:* The mortar paste then was homogeneously poured into the normal moulds (standard dimension, i.e., length × width × height = 150 mm × 150 mm × 150 mm) that were put on the vibroplatform for vibrating compaction (vibrating period 20 minutes, vibrating frequency 45 ± 5 Hz and vibrating amplitude 0.3 mm).

The mortar paste in the normal moulds rested for 24 hours indoor, where the temperature and percentage relative humidity (RH) were 20 ± 5 °C and 75 ± 5%, respectively. The mortar specimens (namely, the cubic ones) that have achieved the initial strength were then unloaded from the normal moulds.

*Curing:* The unloaded mortar specimens were collected into the curing chamber, where the temperature and percentage relative humidity (RH) were 20 ± 5 °C and 90 ± 5%, respectively. Particularly, the cubic specimens were submerged into the tank full of the natural seawater that was also obtained from Changzhi Bay of Zhoushan Seas in order to simulate the marine environment. The curing age for the cubic specimens in the natural seawater was 59 days and the total age, including the resting indoor period, was 60 days. 

*Coring and cutting:* The mortar cores were then drilled from the cubic specimens by corer HZ-18B (Yaxing Ltd., Tianjin, China, power 2.5 kW and maximal rotational speed 1100 r/min.) at 60-day age. Meanwhile, the coarse cores were cut into be the normal cylinder specimens (diameter × height = 50 mm × 100 mm) by cutter ZDQ95-8 (Zhengda Ltd., Linhai, Zhejiang, China) the extreme power and the maximal dimension of the treated specimen of which were 11 kW and length × width × height = 3000 mm × 1200 mm × 150 mm, respectively (Figure 7). 

Furthermore, the normal cylinder specimens were re-collected into the curing chamber and then re-submerged into the tank full of the natural seawater for another 10-day cure, until they achieved 70-day age.

*Tri-axial CT scanner system:* The on-line damage detection experiments have been implemented by the tri-axial CT scanner system (Northwest Institute of Eco-Environment and Resources, CAS, Lanzhou, Gansu, China), which includes two parts, namely, the strain-controlled tri-axial sub-system (Northwest Institute of Eco-Environment and Resources, CAS, Lanzhou, Gansu, China) and the PHILIPS Brilliance 16 MSHCT sub-system (Philips N.V., Amsterdam, Netherlands). Table 5 shows the principal parameters of the tri-axial CT scanner system. 

The axial load $\sigma_{a}$ was executed with the rigid condition by the vertical main-shaft made by DCCr12Mov steel and the principal stress $\sigma_{1}$ was generated in the mortar specimen in the same direction of the axial load. The confining load $\sigma_{c}$ was conducted with the soft condition by the oil pressure that was worked on the HDPE (High Density Polyethylene) jacket (Figure 8), the thickness and the density of which were 1mm and 0.963 g/cm^3^, respectively. Meantime, the principal stress $\sigma_{2}$ was produced in the mortar specimen in the same direction of the confining load.

The term ‘on-line’ indicates that the normal cylinder specimen can be scanned by MSHCT (Philips N.V., Amsterdam, Netherlands) during the tri-axially loading procedure to visually and dynamically detect the damage evolution in it. Therefore, the synchronization of the visualization and metrization can be carried out to investigate the foci development of the mortar while using seawater and sand (Figure 9). 

(T*_i_* on the strain and stress curve was the corresponding position of MSHCT scan of the *i*th time and *n* was the total of MSHCT scan (*n* = 6 in this study). ε and σ represented the SSS of the mortar. $\sigma_{{}_{1}}^{\left(i\right)}$ and $\sigma_{{}_{2}}^{\left(i\right)}$ designated the principal stress states of the *i*th time. Particularly, the strain and stress states (i.e., SSS) that could quantified the damage mathematically and the MSHCT images that could quantified the damage graphically were synchronically collected by the help of the tri-axial CT scanner system.

Moreover, the thickness of the scanned films of the normal cylinder specimen was determined by the resolution adjustment experiments to uniformly be 3 mm (Figure 10).

## 3. Results and Discussion

The completed specimens nowadays were 270 items and the statistical characteristics work of the mortar will be offered in the next part. The SSS was the key approach for foci development investigation and the SSS of the mortar using seawater and sand is shown in Figure 11, where $\varepsilon_{a}$, $\varepsilon_{v}$, and $\varepsilon_{r}$ denoted the axial strain, the volumetric strain, and the radial one, respectively. Hence, the 1^st^, 2^nd^, and 3^rd^ curves interpreted the relations of the strain states on the principal stress $\sigma_{1}$. Moreover, the cyan rectangles on the three curves of Figure 11 represented the positions of MSHCT scan. The sequence of the positions of MSHCT scan was prescribed by the serial number T*_i_* (*i* = 1~6) and the blue arrows along the 1^st^, 2^nd^, and 3^rd^ curves. Meantime, six marginal points were specified on the three curves of Figure 11, namely, A, B, C, D, E, and F, that, indicated by the pink circles, helped to define the critical states of the physical-chemical-mechanical performance of the mortar specimen.

The damage evolution was reconstructed by the MSHCT approach and the results are given in Figure 12, where the two-dimensional behaviors of the damaged mortar were presented dynamically in six MSHCT scan positions, namely, T_1_, T_2_, T_3_, T_4_, T_5_, and T_6_. Meanwhile, the foci development were correspondingly detected, measured, and reconstructed in six MSHCT scan positions.

Particularly, the characteristics of the mortar specimen were introduced according to Figure 11, with the help of the marginal points, interval by interval, as follows.

***Interval 0-A on the curves in Figure 11:*** The value of $\varepsilon_{a}$ in the direction of the axial load $\sigma_{a}$ was 0 at point A, where the driving shaft (i.e., the vertical main-shaft) under the parking state had no motion and the axial load has not been worked. The confining load of 2 MPa (i.e., $\sigma_{c}$ = 2 MPa), by contrast, has been worked on the normal cylinder specimen at the same time in order to stabilize the radial stress state, which demonstrated the storage environment of the material. The priority of the confining load has been accepted by most tri-axial experiments, because it can simulate the original storage environments of the studied materials. The undisturbed performance of the studied materials in the host could thereby be recovered during the tri-axial experiments. Hence, this procedure was defined as stress maintenance. Consequently, the volumetrically contractive strain at point A was 5.09 × 10^−5^, which was produced by the radial deformation, the value of the radially compressive strain of which was −2.59 × 10^−5^. Furthermore, the volumetrically contractive strain of interval 0-A showed the linearity of the mortar specimen. Meanwhile, the original state of the mortar specimen ((a) in Figure 12) expressed that the temporary compression of the parental focus contributed to the linearly volumetric contraction.

***Interval A-B on the curves in Figure 11:*** The radial strain $\varepsilon_{r}$ value of interval A-B (including point A) in the direction of $\sigma_{c}$ was the minus one, which indicated that the radius of the mortar specimen has been compressed to be the shorter one. −2.17 × 10^−5^ was the value of the radially compressive strain of point B on 3^rd^ curve of Figure 11. The maximally absolute value of the minus radial strain at interval A-B was 1.38 × 10^−4^. Meanwhile, the axial strain that was produced by the axially compressive deformation accumulated up to 1.96 × 10^−3^ (at point B on 1^st^ curve of Figure 11) and the maximally volumetric strain of interval A-B attained 1.99 × 10^−3^ (at point B on 2^nd^ curve of Figure 11). Hence, it can be deduced that the totally volumetric deformation here was the contractive one, in both the axial direction and radial direction. However, the minus radial strain with the maximally absolute value and the maximally volumetric strain did not arise at the same time at interval A-B. The maximally volumetric strain lagged behind the minus radial strain, with the maximally absolute value at interval A-B. Furthermore, the development of the axial strain and the volumetric one significantly showed the linear and the non-linear characteristics, respectively. Meanwhile, the radial strain here also developed non-linearly. Consequently, the non-linearly radial strain has caused the non-linearity of the volumetric strain, the contribution to which was the micro-cavities’ compaction in Figure 13. The gel-like constituents around the micro-cavities (including C-S-H and Friedel’s salt) were locked in the radial direction, which created the non-linearly radial strain. Meantime, Figure 13 showed that both the volumetric strain and radial strain were the contractive ones and the C-S-H contraction helped to imprison the local Friedel’s salt. Thereby, the dispersion of Friedel’s salt was impeded.

The generation of the hydrate and the salt in Figure 13 was interpreted by the following equations.
(3)$$2\left({\mathrm{Ca}}_{3}{\mathrm{SiO}}_{5}\right){+6H}_{2}O=3\mathrm{CaO}\cdot {2\mathrm{SiO}}_{2}\cdot {3H}_{2}O+3\mathrm{Ca}{\left(\mathrm{OH}\right)}_{2}$$
(4)$$\mathrm{Ca}{\left(\mathrm{OH}\right)}_{2}+2{\mathrm{NaCl}=\mathrm{CaCl}}_{2}+2{\mathrm{Na}}^{+}+2{\mathrm{OH}}^{-}$$
(5)$${\mathrm{Ca}}_{3}{\mathrm{Al}}_{2}O_{6}+{\mathrm{CaCl}}_{2}+10H_{2}O\to \left({\mathrm{Ca}}_{3}{\mathrm{Al}}_{2}O_{6}\right)\cdot {\mathrm{CaCl}}_{2}\cdot 10H_{2}O$$

***Interval B-C on the curves in Figure 11:*** In terms of the direction of $\sigma_{1}$(i.e., the direction of the axial load $\sigma_{a}$), the peak strength of the mortar using seawater and sand was 33.88 MPa, indicated by point C on the three curves in Figure 11. The values of $\varepsilon_{a}$ and $\varepsilon_{v}$ at point C were 1.23 × 10^−2^ and 1.0 × 10^−2^, respectively. Correspondingly, they accounted for 62.11% and 95.61% of the peak values of $\varepsilon_{a}$ and $\varepsilon_{v}$ (denoted by point D on the 1^st^ and 2^nd^ curves in Figure 11). Meantime, the radial strain at point C in the 3^rd^ curve was 1.09 × 10^−3^. Particularly, the axial strain of the interval B-C was produced by the compressive deformation in the direction of $\sigma_{1}$. On the contrary, the radial strain here was produced by the tensile deformation in the direction of $\sigma_{c}$. Moreover, the volumetric strain of the interval B-C has been generated by the global contraction of the normal cylinder specimen. Therefore, the conclusions that are deduced from the characteristics of the interval B-C included that the development of the peak strength would consume more than half of the axially compressive strain and most of the volumetrically contractive strain; however, the radial strain with the soft loading condition was under-developed when the mortar reached its peak strength in the $\sigma_{1}$ direction; the radial strain here, at the same time, indicated the absolutely tensile deformation in the $\sigma_{c}$ direction; hence, the axial compression has dwarfed the radial extension and resultantly generated the global contraction of the normal cylinder specimen. Meanwhile, the MSHCT state of the mortar specimen ((b) in Figure 12) showed that the distribution of the cohesive continuum around the parental focus has been explicitly disturbed and the promising damage zones have been fused to be a connectively larger area in the core of which the outline of the parental focus was squeezed to be a smooth configuration. Moreover, a new baby focus was generated at the 12 o’clock direction of the parental one in position T_3_. The distance between the parental focus and the new baby one was 11.31mm and their connective region was destroyed to be the macro-fragments ((c) in Figure 12). The maximal size of the macro-fragments reached 4 mm at interval B-C.

***Interval C-D on the curves in Figure 11:***$\varepsilon_{a}$ and $\varepsilon_{v}$ reached their peak values (as indicated by point D on the 1^st^ and 2^nd^ curves) at the same stress state where $\sigma_{1}$ = 20.34 MPa at the backbone of the hysteresis loop in Figure 11. The peak values of $\varepsilon_{a}$ and $\varepsilon_{v}$ were 1.61 × 10^−2^ and 1.14 × 10^−2^, respectively. Meantime, the mortar strength began to drop steeply after the peak value (represented by point C on the three curves in Figure 11) towards the strain coordinate axis. On the other side, the axially compressive strain of interval C-D still increased along the negative direction of the $\sigma_{1}$ coordinate axis. Similarly, the radial strain here that was caused by the tensile deformation showed the reversely climbing trend when compared with the radial strain development of interval B-C. In contrast, in the position T_5_ of 2^nd^ curve, a locally odd fall lived, which indicated the relatively dilative deformation. Furthermore, the locally high-angle zones happened directly in position 6 of both the axial strain curve and the radial strain one (namely, the 1^st^ and 3^rd^ curves in Figure 11). Especially, the locally odd fall on the 2^nd^ curve triggered the fluctuation of the volumetric strain around it, which denoted the alternate deformations of the volumetric contraction and dilation. The maximal values of the contractive strain and the dilative one of volumetric deformation here were 1.03 × 10^−^^2^ and 4.02 × 10^−4^, respectively. Therefore, it can be confirmed that the local fractures that presented the abruptly volumetric deformation have burst in the normal cylinder specimen ((d) in Figure 12); the locally steep rise of the radially tensile strain defeated the development of the axially compressive strain, which created the relatively dilative deformation of the mortar (position T_5_ of 2^nd^ curve and (e) in Figure 12). The local fracture that caused the abrupt volumetric dilation was built by three zones, namely: the interface failure zone, where the sea sand particle was pulled half out of the hydrate layers and the framework of the mortar was collapsed; the crack zone, where the hydrate layers, including mainly C-S-H, C-A-S-H, and N-A-S-H gel were partially broken into a volumetric crack; and, the breakage zone, where the needle-like AFt (namely, ettringite) was snapped and its clastic ones clogged and enlarged the volumetric crack with the coupled effects of its growth and the load accumulation (Figure 14).

Especially, the perimeter of the sea sand particle that was pulled out of half the hydrate layers was covered with the salt chips which precipitated the interfacial damage in the crack zone due to the shortage of smaller sea sand particles that could help to absorb the salt chips.

***Interval D-E on the curves in Figure 11:*** The axial strain $\varepsilon_{a}$ began to drop after the peak value 1.61 × 10^−2^ at point D of the 1^st^ curve in Figure 11. The principal stress $\sigma_{1}$ at point E of the three curves in Figure 11 was 13.53 MPa, with the correspondingly axial strain 1.5 × 10^−2^, where the framework of the normal cylinder specimen began to crumble due to the re-adjustment of $\varepsilon_{a}$ development. Particularly, the re-adjustment of $\varepsilon_{a}$ development produced the loose connection among the macro-fragments and undermined the mortar strength. Meanwhile, the smart extensometer of the tri-axial sub-system decelerated tracing the axial deformation to prohibit the possible injury against the sub-system. The driving shaft then started to gradually rest here. Hence, coupled with the strength reduction, the dropping trend of the axially compressive strain represented that the axially tensile deformation was marginally recovered as a result of the gradual rest of the vertical main-shaft, which helped to partially retrieve the elasticity of the mortar. Moreover, the confining load here failed to effectively confine the radially tensile deformation due to the crumbled framework of the normal cylinder specimen. The maximal gradient of the radial tensile strain with the axial load (i.e., $\varepsilon_{r}$/$\sigma_{a}$) resided at interval D-E and the value was 4.03 × 10^−4^/MPa that indicated the relatively volumetric dilation of the mortar. The 2^nd^ curve in Figure 11 showed that the maximal value of the relatively volumetric dilative strain at interval D-E was 4.94 × 10^−3^, which denoted that the confining load could not resist the volumetric dilation in radial direction again. Particularly, the partially retrieved elasticity in the axial direction also contributed to the relatively volumetric dilation of the mortar the framework of which has been radically re-organized during the elastic recovery.

***Interval E-F on the curves in Figure 11:*** The mortar strength was softened by the penetrating cracks ((f) in Figure 12) at interval E-F and its framework absolutely collapsed due to the widespread microscopic deterioration in Figure 15. 

The generation of AFt and feldspar mixture formed the well-blockage in the mortar specimen where the hydration was locally retarded and the discontinuous foci were created under the loading conditions. The discontinuous foci at the initially loading stage were supported by the stick-like elements of AFt and feldspar mixture. Eventually, the elements were radically crashed into chips and the local hydrate that was the constraint boundary of the foci at the initially loading stage was collapsed due to the fatigue damage that caused the evolution of the baby cracks. Therefore, the foci neighborhood thoroughly deteriorated with loss of the bracement from the local hydrate and AFt and feldspar mixture. Figure 16 depicts the procedure on the widespread microscopic deterioration in the mortar using seawater and sand, as follows.

The focus and its constraint from the local hydrate and AFt and feldspar mixture formed the hyper-static framework under the initial state. The moments in focus under the initial state were the lower ones and the microscopic deterioration in the mortar did not spread yet ((1) in Figure 16). Meantime, the moments here could be defined by the parabolic functions with the postulation that the uniform load worked on the focus the volume of which was the infinitesimal one. However, the focus failed to be solidified with its constraint as the result of the discontinuousness that blocked the locally advanced hydration. The cumulative load broke the stick-like elements of AFt and feldspar mixture that were isolated from the hydration. Therefore, the framework that was formed by the focus and its constraint became the statically determinate one with the higher moment ((2) in Figure 16). Thereby, the focus was damaged to be the baby ones that were loaded with the higher moment. Meantime, the local hydrate that was the constraint boundary of the baby foci was also disturbed ((3) and (4) in Figure 16). Moreover, the baby foci with their constraint always formed the unreliable framework of the statically determinate one from then on, and the breach of the framework kept going due to the higher moments until the stick-like elements were radically crashed into chips. The foci then had to stand on the chips and sustained the infinitely concentrated load, which could be expressed by the Dirac function in Equation (6). Consequently, the microscopic deterioration spread out of control ((n) in Figure 16).
(6)$$\delta \left(x\right)=\{\begin{array}{c} \frac{1}{2\xi }\left|x\right|\leq \xi \\ 0\left|x\right|>\xi \end{array}$$
where *x* denoted the possibly loading domain; and, ξ represented the foci neighborhood that approached the infinitesimal. Hence, the nano-scale foci can easily deteriorate due to the infinite load of δ(x).

Particularly, the resilient performance of the axial load at interval E-F did not mean that the strength of the mortar was retrieved. Coupled with the unconfined extension in the radial direction (3^rd^ curve in Figure 11), the re-increment of the axial strain (1^st^ curve in Figure 11) indicated that the volumetric dilation of the mortar was out of control, with which, the resilient performance of the axial load caused the absolute collapse of the framework instead of the strength retrieve. The residually axial load $\sigma_{a}$ in the position F was 13.64 MPa. Correspondingly, the axial strain $\varepsilon_{a}$, the volumetric strain $\varepsilon_{v}$, and the radial one $\varepsilon_{r}$ here were 1.52 × 10^−2^, 5.19 × 10^−3^, and 4.95 × 10^−3^, respectively. Moreover, the residual strength could help prevent sharp destruction of engineering bodies that are composed with the mortar using seawater and sand.

## 4. Conclusions

In this paper, the on-line damage detection experiments were implemented to study the physical-chemical-mechanical characteristics of the mortar using seawater and sand. The foci development in the mortar specimen was then evaluated. The following conclusions based on them can be drawn:
The main source of the saline minerals in the mortar was the seawater used for the specimens’ preparation. The saline minerals were partly dissipated as a result of the curing treatment. However, the residual saline minerals generated the micro-interfacial flaws coupled with the uniform and discontinuous distribution of the sea sand particles. The local fracture could be controlled by the active adjustment for the distribution of the sea sand. The bigger sea sand particles could strengthen the framework of the mortar and the smaller ones played the key role in residual saline minerals’ absorption. The micro-interfacial behavior of the sea sand and the hydrate gel partially controlled the performance of the mortar. The factors inclusive of the curing, the surface pattern of the sea sand, and the gel type crucially worked on the micro-interfacial behavior, which should be specifically studied.The principal constituents of the composite Portland cement P.C 42.5 R included Ca_2_SiO_4_, Ca_3_SiO_5_, and Ca_3_Al_2_O_6_ that contributed to the hydrate generation of the mortar using seawater and sand.The cause why the on-line damage detection experiments art was designed is that the art helps to quantify the foci development by the visualized measurement of the damage. According to the results from the on-line damage detection experiments, the priority of the confining load produced the linearly volumetric contraction due to the temporary compaction of the parental foci in the mortar specimen. Particularly, the radially compressive deformation was the only contribution to the linearly volumetric contraction during stress maintenance. With the birth of the axial strain, the gel-like constituents around the micro-cavities began to be softened, which created the non-linearly radial strain. Meanwhile, the non-linearly radial strain triggered the development of the non-linearly volumetric deformation of the mortar while using seawater and sand. The mortar specimen was scanned for twice besides the original state scan before the peak strength, which was 33.88 MPa. The development of $\varepsilon_{a}$ and $\varepsilon_{v}$ here accounted for more than half of their peak values, by which the kneaded parental focus gave birth to a baby one. Consequently, the connective region between them was destroyed to be the fragments due to the resultantly global contraction.$\varepsilon_{a}$ and $\varepsilon_{v}$ achieved their peak values at the same stress state, where their values were 1.61 × 10^−2^ and 1.14 × 10^−2^, respectively. There lived between the peaks of $\varepsilon_{a}$ (or $\varepsilon_{v}$) and $\sigma_{1}$ the locally high-angle zones of the SSS curves and the local fractures were produced right here. Particularly, the abruptly volumetric dilation was caused by the local fractures that were composed of the interface failure zone of SiO_2_, the crack zone of the hydrate layers, and the breakage zone of the needle-like AFt. Moreover, the radial tensile strain met a sharp knee corner right in the position where $\varepsilon_{a}$ and $\varepsilon_{v}$ achieved their peak values. The $\varepsilon_{r}$ development dashed out of control afterwards. Coupled with the maximal gradient of the radial tensile strain with the axial load, the partially retrieved elasticity in the axial direction caused the relatively volumetric dilation of the mortar using seawater and sand.Upon the penetrating cracks’ birth, the neighborhood of the parental foci was thoroughly deteriorated, where the widespread microscopic failure destroyed the reliable support from the local hydrate and the mixture of AFt and feldspar. Consequently, the parental foci with the discontinuousness and the well-blockage effect triggered the Domino damage in the mortar specimen.The development of $\varepsilon_{a}$, $\varepsilon_{v}$, and $\varepsilon_{r}$ achieved the coupled effect in the mortar performance, although their behaviors showed the different characteristics. The synchronization of SSS metrization and the damage visualization established in this paper was the crucial approach for research on the physical-chemical-mechanical characteristics of the mortar using seawater and sand and cured in natural seawater.

## Figures and Tables

**Figure 1 materials-12-01799-f001:**
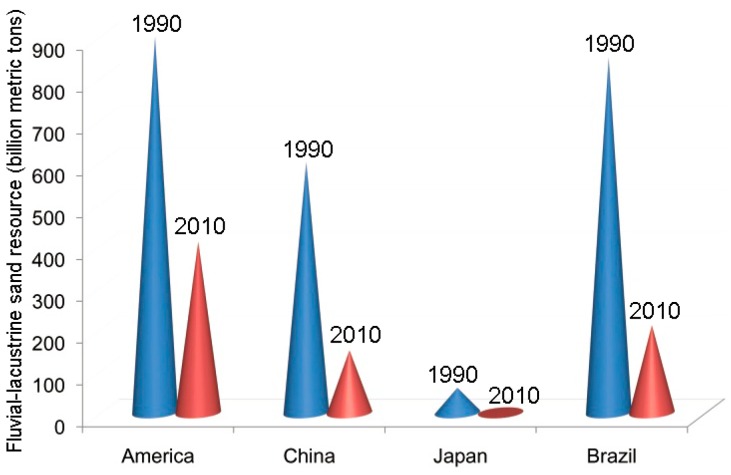
World fluvial-lacustrine sand resource [3].

**Figure 2 materials-12-01799-f002:**
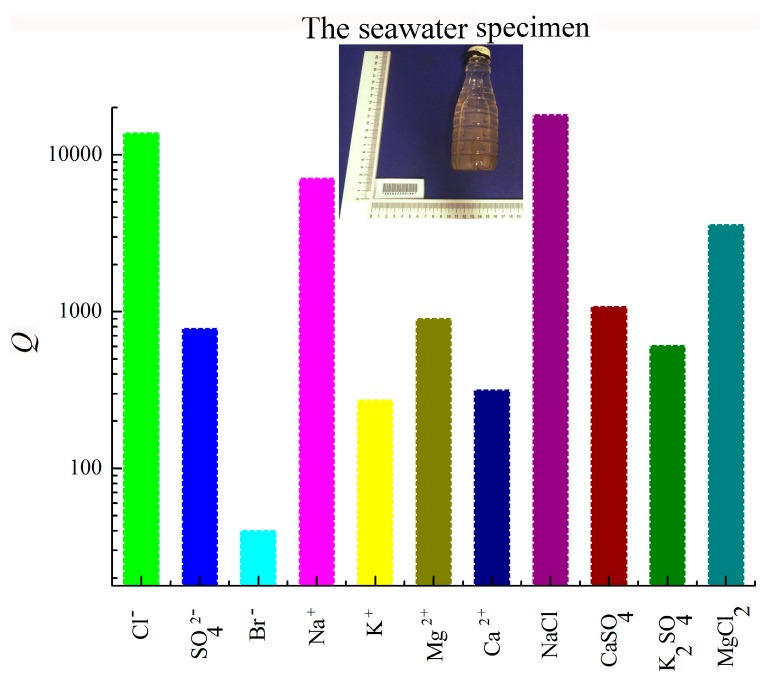
Quantitative distribution of the key ions and the saline minerals of the seawater specimen (The unit of *Q*, namely, the quantification, is 10^−3^·‰).

**Figure 3 materials-12-01799-f003:**
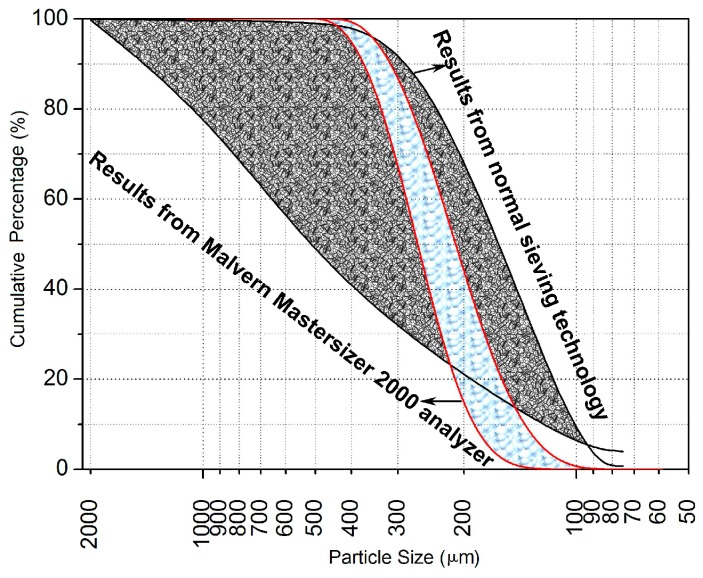
Size distribution of the sea sand.

**Figure 4 materials-12-01799-f004:**
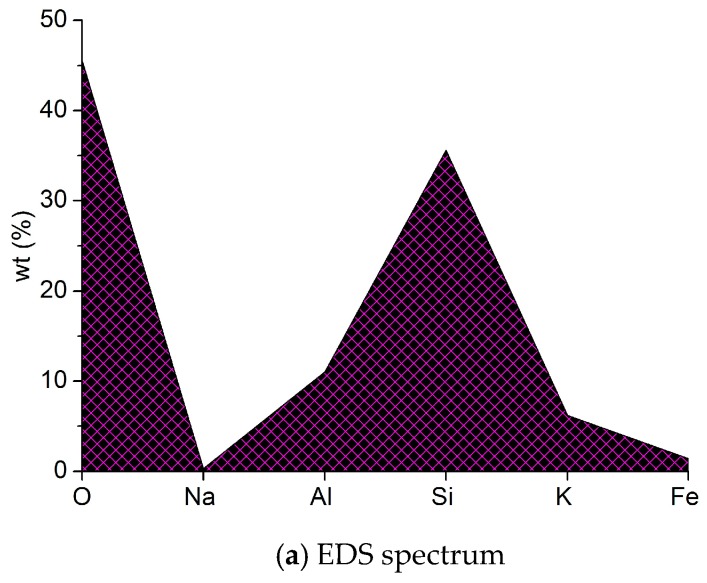
EDS spectrum and XRD patterns of the sea sand.

**Figure 5 materials-12-01799-f005:**
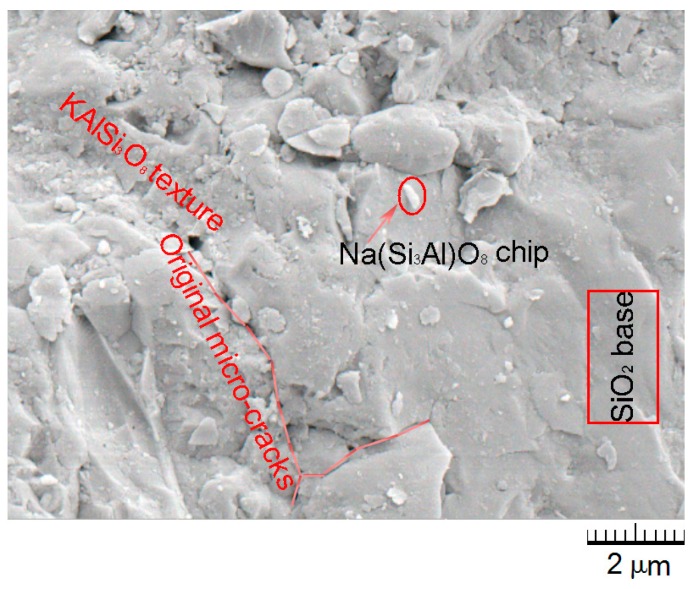
SEM image of the sea sand.

**Figure 6 materials-12-01799-f006:**
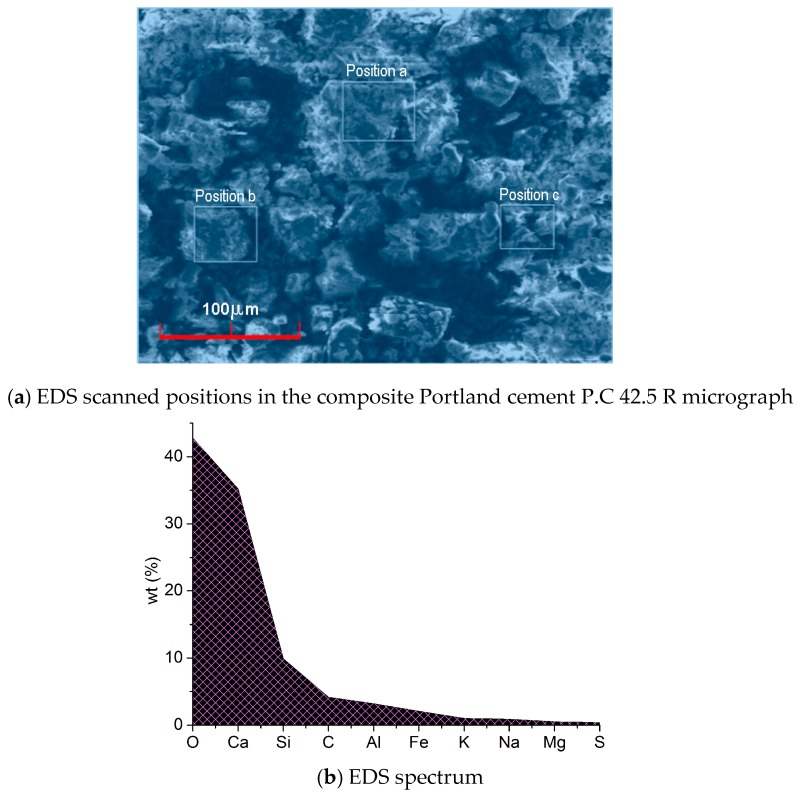
Micrograph and principal elements’ content from EDS analysis on the composite Portland cement P.C 42.5 R.

**Figure 7 materials-12-01799-f007:**
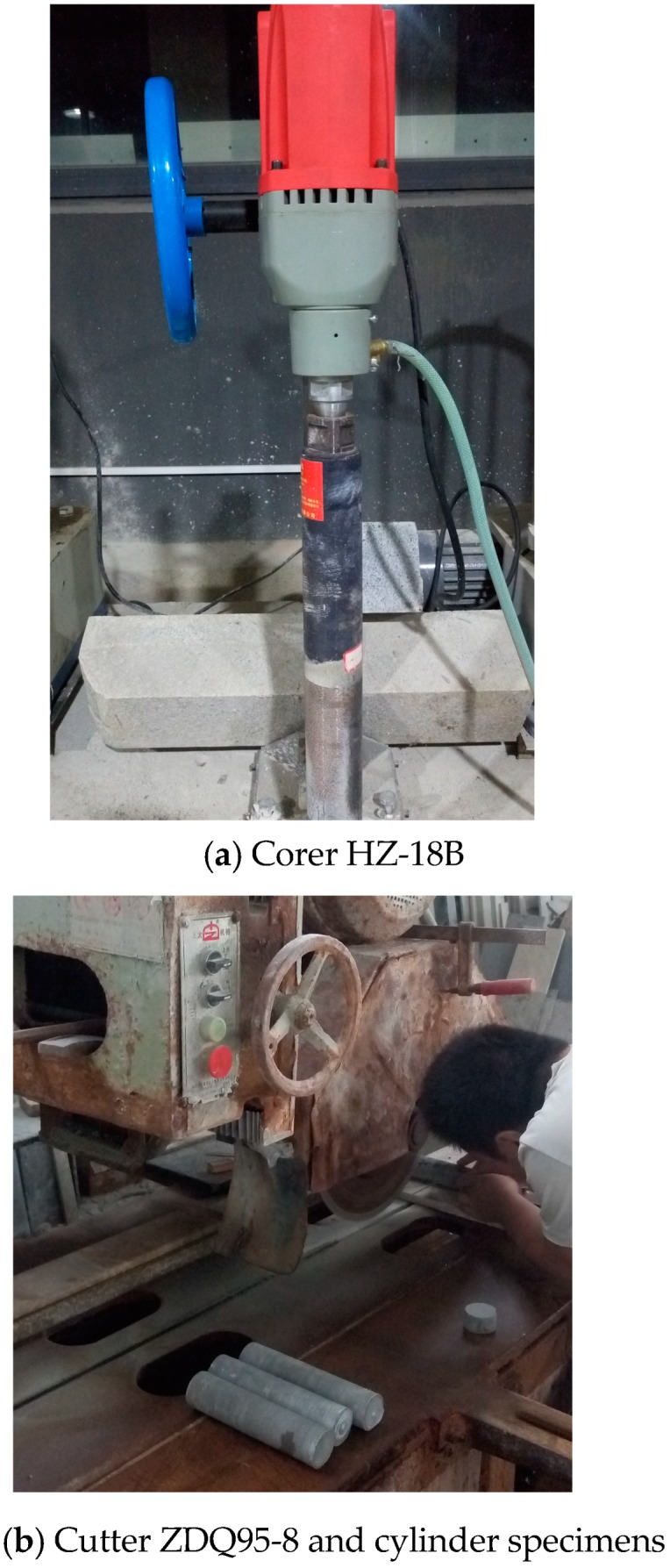
Production of the normal cylinder specimens.

**Figure 8 materials-12-01799-f008:**
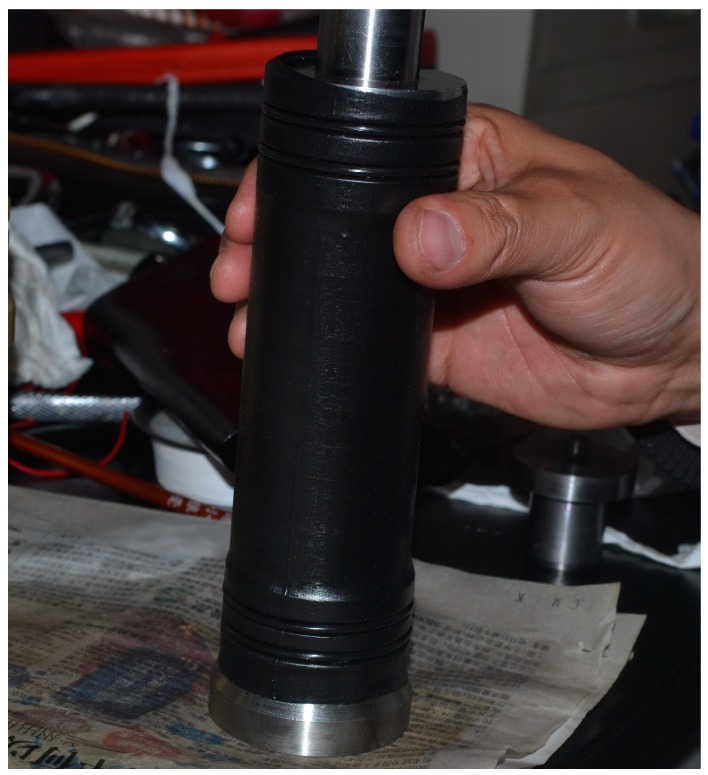
Normal cylinder specimen covered with High Density Polyethylene (HDPE) jacket.

**Figure 9 materials-12-01799-f009:**
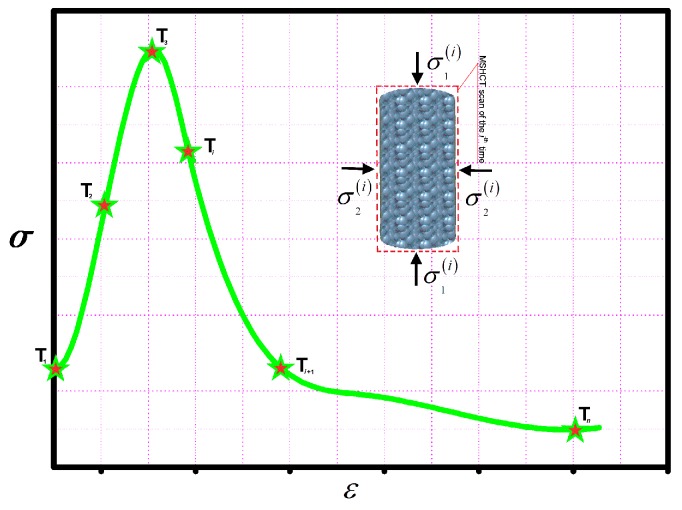
Synchronization of the visualization and metrization on the foci development.

**Figure 10 materials-12-01799-f010:**
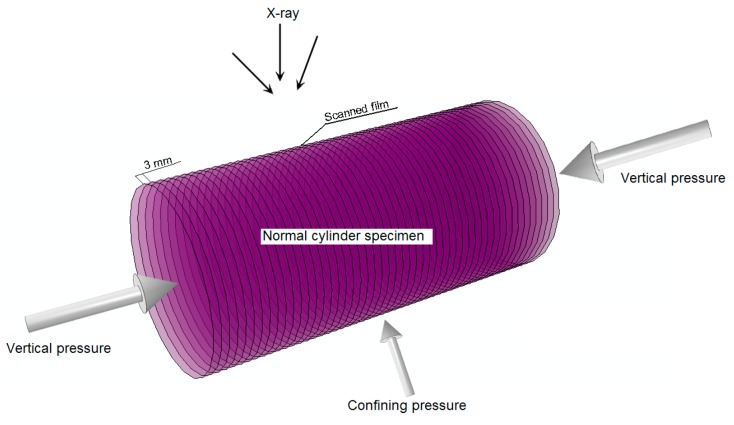
On-line damage detection model.

**Figure 11 materials-12-01799-f011:**
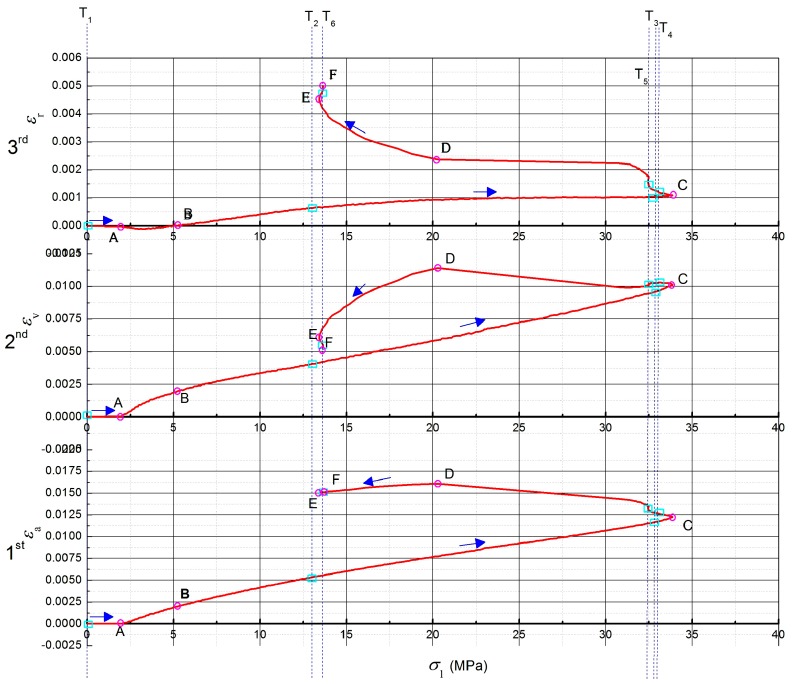
Strain and stress states (SSS) of the mortar specimen.

**Figure 12 materials-12-01799-f012:**
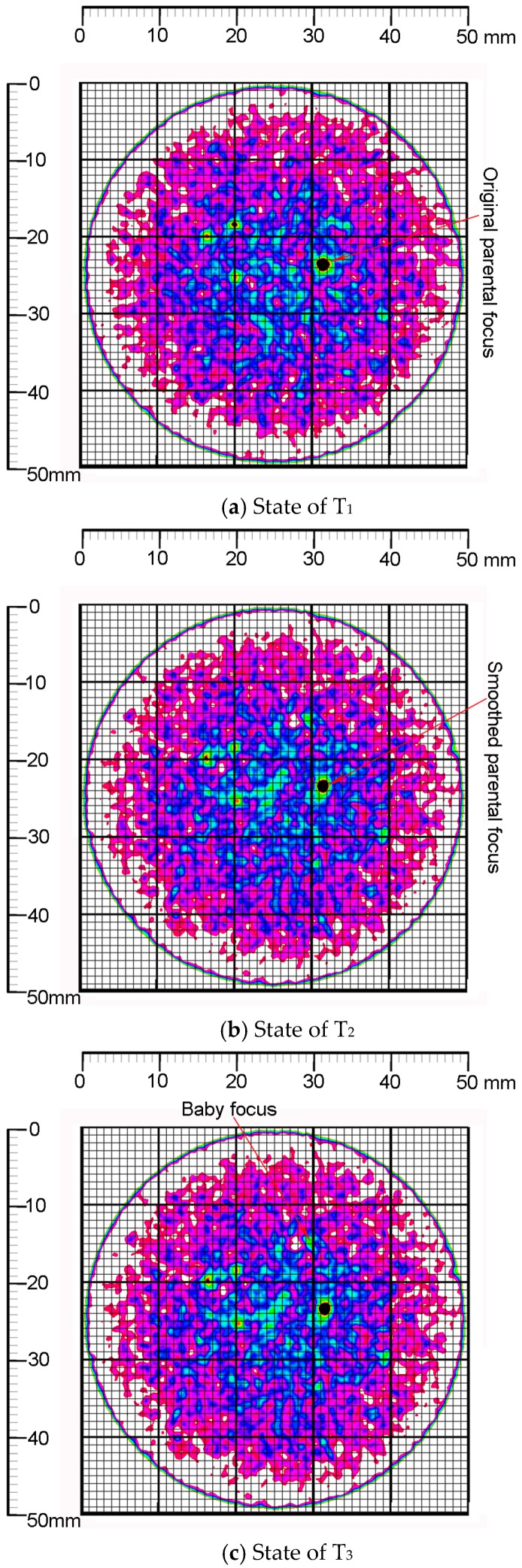
Reconstructed models of damage evolution of the normal cylinder specimen.

**Figure 13 materials-12-01799-f013:**
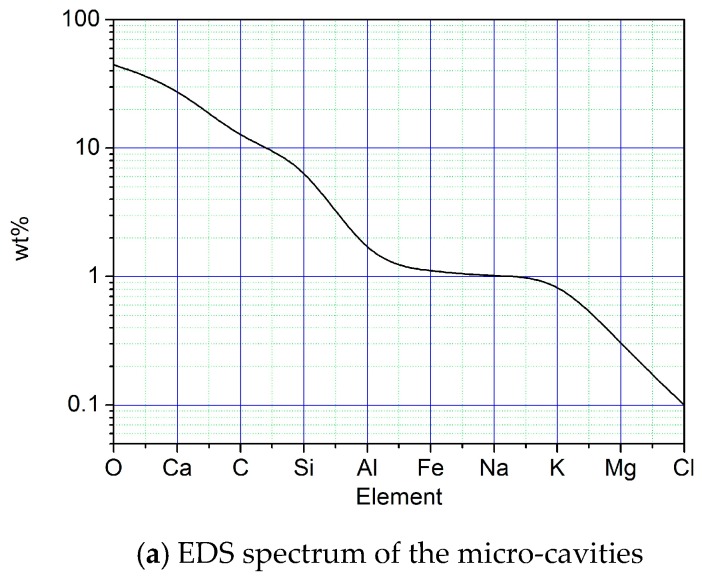
Behaviors of micro-cavities in the mortar using seawater and sand.

**Figure 14 materials-12-01799-f014:**
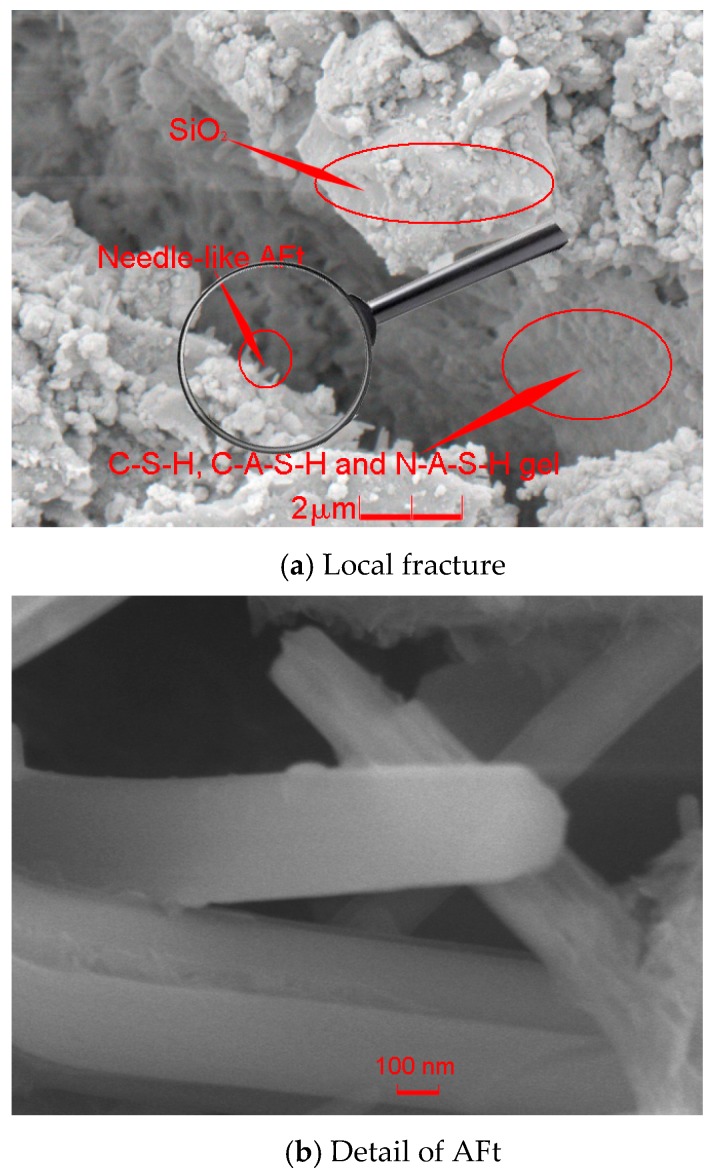
SEM image of the local fracture and volumetric dilation.

**Figure 15 materials-12-01799-f015:**
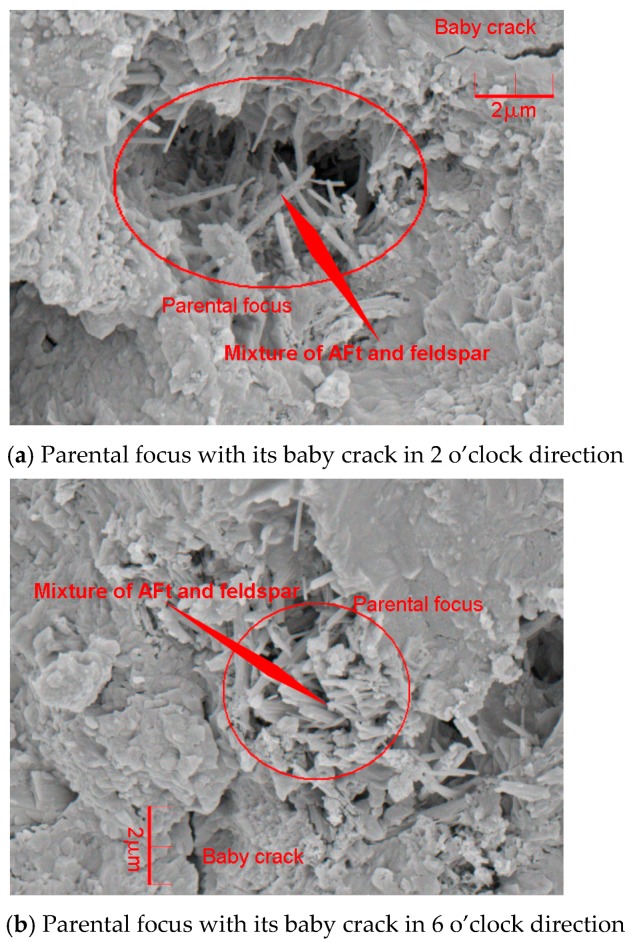
SEM images of microscopic deterioration due to the foci development.

**Figure 16 materials-12-01799-f016:**
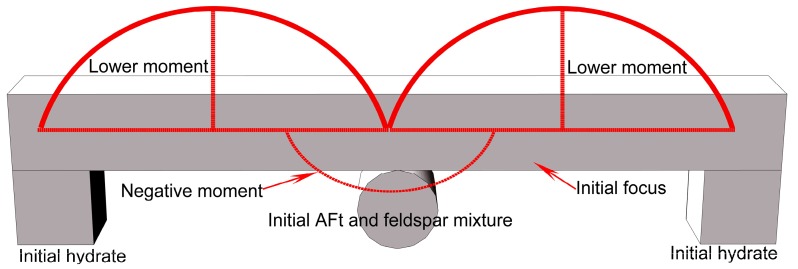
Widespread microscopic deterioration model.

**Table 1 materials-12-01799-t001:** Composition of the mortar.

Material	Amount (g)
Cement	1061
Sea sand	803
Seawater	486 (W/C = 0.458)

**Table 2 materials-12-01799-t002:** Principal properties of the sea sand.

Apparent Density (kg/m³)	Packing Density (kg/m³)	Natural Water Content (%)	Methylene Blue Number (‰)	Shell Residuals Contents (%)	Sludge Contents (%)
2652	1370.3	20%	1.4	0.23	0.27

**Table 3 materials-12-01799-t003:** Expectation values of the sea sand characteristic parameters.

Results from Normal Sieving Technology						Results from Malvern Mastersizer 2000 Analyzer					
*d*_10_(μm)	*d*_30_(μm)	*d*_50_(μm)	*d*_60_(μm)	*C* _u_	*C* _c_	*d*_10_(μm)	*d*_30_(μm)	*d*_50_(μm)	*d*_60_(μm)	*C* _u_	*C* _c_
1296	1615	2380	3054	2.36	0.66	162.617	203.008	238.943	256.571	1.58	0.99

**Table 4 materials-12-01799-t004:** Principal properties of the composite Portland cement P.C 42.5 R.

Initial Setting Time (minute)	Final Setting Time(minute)	Apparent Density (kg/m³)	Compressive Strength for 28 Days (MPa)
100	167	3200	55.8

**Table 5 materials-12-01799-t005:** Principal parameters of tri-axial CT scanner system.

**(a) Tri-Axial Sub-System**																								
**Outline Dimensions (mm)**				**Tri-Axial Cabinet**					**Extreme Loading Conditions**							**Maximal Stroke of Vertical Main-Shaft (mm)**			**Accuracies (%)**					
				**Diameter** **(mm)**				**Height** **(mm)**	**Force from Vertical Main-Shaft (kN)**					**Confining Pressure** **(MPa)**					**Deformation Controller**				**Loading Controller**	
Height: 700 Maximal diameter: 305 Minimal diameter: 225				50				100	500					20		150			0.05				0.1	
**(b) MSCT sub-system**																								
**Gantry mouth diameter** **(mm)**		**Coverage width of detector axis Z (mm)**			**Gantry gradient**		**Detector material**			**Detectors sum**		**Detector channels sum**			**Detector cooling method**		**Scanning space accuracy** **(mm)**	**Message transmission rate (GB/s)**		**X-ray generator**				
																				**Power (kW)**	**Maximal amperage (mA)**			**Maximal voltage (kV)**
700		24 mm			±30°± 0.5°		Express rare earth ceramics			672		1344			Air cooling one		0.75	1.1		60	500			140
**(c) MSCT sub-system**																								
**X-ray generator**						**Contrast resolution (%)**					**Spatial resolution** **(mm)**		**Image reconstruction matrix dimension**			**CT value span**			**Contrast resolution**			**Spatial resolution** **(mm)**		
**Power** **(kW)**	**Maximal amperage (mA)**		**Maximal voltage** **(kV)**																					
60	500		140			0.3					0.2		1024 × 1024			[−1024, 3071]			0.3%			0.2		

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
