# Peer review of "Experimental Study on Foci Development in Mortar Using Seawater and Sand"

_materials, 2019, doi:10.3390/ma12111799_

Round 1
Reviewer 1 Report
The paper "Multi-Scale Physical-Chemical_Mechanical Characteristics of Marine Mortar part I -Primary Performance" is very well written and complete. It discuss on several aspects of interest in the field of sea water and sea sand in the production of cement paste. In the specific Figure 2 is stressing about the importance of this kind of study. I only suggest to eliminate some useless figures (Figure 5, 11 and 12; choose one between Figure 13 b and c); to verify if any of the XRD patterns shown in Figure 10 is needed; to verify the format of Table 5.
Please verify the format of the Font not visible at line 115 (not visible in the pdf file)
Please add a comma in line 307: ...radial strain, contribution to which....
Please rearrange the format of Figure 22; in the current state the colors matching does not allow a good legibility.
Author Response
1. I only suggest to eliminate some useless figures (Figure 5, 11 and 12; choose one between Figure 13 b and c);
Answer: We accepted absolutely the reviewer’s suggestion and the eliminated figures included Figure 5, 11, 12 and Figure 13 c.
2. to verify if any of the XRD patterns shown in Figure 10 is needed;
Answer: Thanks for the inspiring comments from the reviewer. The XRD patterns of composite Portland cement P.C 42.5 R in Figure 10 have been removed all to stress about the key work of this manuscript. Meantime, the main results of XRD analysis were presented by a short statement in the manuscript.
3. to verify the format of Table 5.
Answer: The format of Table 5 was rectified according to the template of Materials.
4. Please verify the format of the Font not visible at line 115 (not visible in the pdf file)
Answer: We have rectified the format of the Font at line 115 for the fine printing.
5. Please add a comma in line 307: ...radial strain, contribution to which....
Answer: We agreed with the reviewer and the comma has been added at the proper position.
6. Please rearrange the format of Figure 22; in the current state the colors matching does not allow a good legibility.
Answer: We have rearranged the format of Figure 22 to allow a good legibility.

Reviewer 2 Report
A) The introduction part is incoherent, exemplary:
- “The worldwide production of the cement-base materials is skyrocketing (Figure 1).” – The figure one shows the cement production in 2018, it is not connected with any changes.
- “Guo et al. [20] 71 conducted the experimental investigation on the degradation of the fiber reinforced polymer…” - It is not connected with the topic of the article.
- “According to the references, it can be drawn that the marine cement-base composites have been the most popular materials in the offshore construction.” – Lack of information about particular source.
B) The organization of the article requires to be re-arrangement, especially part 2 (sub-chapters).
C) Lack of information about the amount of the specimens.
Author Response
1. A) The introduction part is incoherent, exemplary:
Answer: We agree absolutely with the reviewer and have revised the mentioned parts as follows. We also re-organized the introduction correspondingly with the updated data, figure, reference and statement.
2. - “The worldwide production of the cement-base materials is skyrocketing (Figure 1).” – The figure one shows the cement production in 2018, it is not connected with any changes.
Answer: We understood the comment of the reviewer and updated the data in Figure 1.
3. - “Guo et al. [20] 71 conducted the experimental investigation on the degradation of the fiber reinforced polymer…” - It is not connected with the topic of the article.
Answer: We were highly grateful to the reviewer for the considerate suggestion. We also substituted the proper reference in the manuscript for the old one.
4. - “According to the references, it can be drawn that the marine cement-base composites have been the most popular materials in the offshore construction.” – Lack of information about particular source.
Answer: We accepted the suggestion of the reviewer and included the abundant information and references in the revised manuscript which now can show the particular source.
5. B) The organization of the article requires to be re-arrangement, especially part 2 (sub-chapters).
Answer: The authors re-arranged the article content, especially part 2 (sub-chapters), the organization and format of which now met the requirements of Materials.
11. C) Lack of information about the amount of the specimens.
Answer: Thanks for the helpful comment from the reviewer and we have completed nowadays 270 specimens and the statistical work on their characteristics will be offered in the next part. Surely, this information was also added in the manuscript.

Round 2
Reviewer 2 Report
Required changes have been made.
Author Response
Answers: We accepted absolutely the reviewer’s suggestion. We understood that the results as the important part needed more distinct presentation. The modifications on the results in detail were then mentioned one by one as follows.
1. The paragraph in lines 245 and 246 may be obscure and failed to explain the function of Figure 13. Hence, the old expression was revised precisely as “Hence, the 1st, 2nd and 3rd curves expressed the relations of , and on the principal stress.”
2. The number of Figure 14 in line 247 should be replaced by 13 that indicated the modified figure.
3. The long sentence in line 247 was split into two short sentences that can express the results distinctly. The revision was “Moreover, the cyan rectangles on the three curves of Figure 13 represented the positions of MSHCT scan. The sequence of the positions of MSHCT scan was prescribed by the serial number Ti (i =1~6) and the blue arrows along the 1st, 2nd and 3rd curves.”
4. The preposition “at” in line 256 was replaced properly by “in”.
5. The long sentence in lines 256 and 257 was split into two short sentences that can express the results distinctly. The revision was “The on-line multi-scale damage evolution was reconstructed by MSHCT approach and the results were given in Figure 14 where the 2-dimensional behaviors of the damaged marine mortar were presented dynamically in 6 MSHCT scan positions, namely, T1, T2, T3, T4, T5 and T6. Meanwhile, the correspondingly intrinsic fractures were detected, measured and reconstructed in 6 MSHCT scan positions.”
6. The long sentence in lines 278 and 279 was split into two short sentences that can express the results distinctly. The revision was “The undisturbed performance of the goal materials in the host could be recovered during the tri-axial experiments”.
7. The number of Figure 18 in line 285 should be replaced by 14 that indicated the modified figure.
8. The sentence in lines 289, 290 and 291 was modified as “-2.17×10-5 was the value of the radially compressive strain of point B on 3rd curve of Figure 13. The maximally absolute value of the minus radial strain at interval A-B was 1.38×10-4”. The new expression can depict the results more distinctly.
9. The updated expression on the minus radial strain was added in line 295 to show the results clearly as “However, the minus radial strain with the maximally absolute value and the maximally volumetric strain did not arise at the same time at interval A-B. The maximally volumetric strain lagged behind the minus radial strain with the maximally absolute value at interval A-B”.
10. The preposition “at” in line 338 was replaced properly by “in”.
11. The statement in lines 344, 345 and 346 was updated as “Meantime, the strength of the marine mortar began to drop steeply after the peak strength (represented by point C on the three curves in Figure 13) towards the strain coordinate axis” where the adverb was adjusted.
12. The statement in line 349 was polished as “Similarly, the radial strain here that was caused by the tensile deformation showed the reversely climbing trend when compared with the radial strain development of interval B-C” the expression of which was more precise.
13. The preposition “at” in line 349 was replaced properly by “in”.
14. The statement in lines 350, 351 and 352 was modified as “Furthermore, the locally high-angle zones happened directly in the position 6 of both the axial strain curve and the radial strain one (namely, the 1st and 3rd curves in Figure 13)” the expression of which was more precise.
15. The preposition “at” in line 357 was removed to state the information in Figure 14 concisely.
16. The sentence in line 364 was polished to express the local fracture characteristics more distinctly.
17. The long sentence in lines 373, 374,375 and 376 was split into two short sentences that can express the results distinctly. The revision was “The axial strain began to drop after the peak value 1.61×10-2 at point D of the 1st curve in Figure 13. The principal stressat point E of the three curves in Figure 13 was 13.53 MPa with the correspondingly axial strain 1.5 ×10-2 where the framework of the normal cylinder specimen began to crumble due to the re-adjustment of development”. The new sentences can depict the SSS more precisely.
18. The long sentence in lines 378, 379 and 380 was re-written as “Meanwhile, the smart extensometer of the tri-axial sub-system decelerated tracing the axial deformation to prohibit the injury against the sub-system. The driving shaft then started to rest here gradually.”
19. The sentence in lines 380, 381, 382 and 383 was modified as “Hence, coupled with the strength reduction, the dropping trend of the axially compressive strain represented that the axially tensile deformation was recovered marginally as a result of the gradual rest of the vertical main-shaft which helped retrieve partially the elasticity of the marine mortar” which can express more definitely the strength evolution of the marine mortar.
20. The sentence in lines 383 and 384 was re-written as “Moreover, the confining load here failed to confine effectively the radially tensile deformation due to the crumbled framework of the normal cylinder specimen” which was the more thoughtful one.
21. The long sentence in lines 424, 425 and 426 was split into two short sentences as “The focus and its constraint from the local hydrate and AFt and feldspar mixture formed the hyper-static framework under the initial state. The moments in the focus under the initial state were the lower ones and the microscopic deterioration in the marine mortar did not spread yet ((1) in Figure 18)”. The new statements expressed the focus mechanism more clearly.
22. The long sentence in lines 430, 431 and 432 was split and re-written as “The cumulative load broke the stick-like elements of AFt and feldspar mixture that were isolated from the hydration. Therefore, the framework formed by the focus and its constraint became the statically determinate one with the higher moment ((2) in Figure 18)”. The re-written sentences helped explain the focus development more specifically.
23. The long sentence in lines 433, 434 and 435 was split into two clauses that showed distinctly the widespread microscopic deterioration.
24. The preposition “at” in line 450 was replaced properly by “in”.
